# Uakitite, VN, a New Mononitride Mineral from Uakit Iron Meteorite (IIAB) †

**Victor V. Sharygin** [1,2,3,*], **German S. Ripp** [4], **Grigoriy A. Yakovlev** [3], **Yurii V. Seryotkin** [1,2], **Nikolai S. Karmanov** [1], **Ivan A. Izbrodin** [4], **Victor I. Grokhovsky** [3] and **Elena A. Khromova** [4]

1  V.S. Sobolev Institute of Geology and Mineralogy, Siberian Branch of the RAS, 3 Acad. Koptyuga pr., 630090 Novosibirsk, Russia; yuvs@igm.nsc.ru (Y.V.S.); krm@igm.nsc.ru (N.S.K.)
2  Department of Geology and Geophysics, Novosibirsk State University, 2 Pirogov str., 630090 Novosibirsk, Russia
3  ExtraTerra Consortium, Institute of Physics and Technology, Ural Federal University, 19 Mira str., 620002 Ekaterinburg, Russia; yakovlev.grigoriy@gmail.com (G.A.Y.); grokh47@mail.ru (V.I.G.)
4  Geological Institute, Siberian Branch of the RAS, 6a Sakhyanovoi str., 670047 Ulan-Ude, Russia; ripp@ginst.ru (G.S.R.); izbrodin@ginst.ru (I.A.I.); lena.khromova.00@mail.ru (E.A.K.)
*  Correspondence: sharygin@igm.nsc.ru; Tel.: +7-383-330-80-84
†  The paper was presented at the 81st Annual Meeting of the Meteoritical Society in Moscow, Russia, 22–27 July 2018.

**Abstract:** Uakitite was observed in small troilite–daubréelite (±schreibersite) inclusions (up to 100 μm) and in large troilite–daubréelite nodules (up to 1 cm) in Fe-Ni-metal (kamacite) of the Uakit iron meteorite (IIAB), Republic of Buryatia, Russia. Such associations in the Uakit meteorite seemed to form due to high-temperature (>1000 °C) separation of Fe-Cr-rich sulfide liquid from Fe-metal melt. Most inclusions represent alternation of layers of troilite and daubréelite, which may be a result of solid decay of an initial Fe-Cr-sulfide. These inclusions are partially resorbed and mainly located in fissures of the meteorite, which is now filled with magnetite, and rarely other secondary minerals. Phase relations indicate that uakitite is one of the early minerals in these associations. It forms isometric (cubic) crystals (in daubréelite) or rounded grains (in schreibersite). The size of uakitite grains is usually less than 5 μm. It is associated with sulfides (daubréelite, troilite, grokhovskyite), schreibersite and magnetite. Carlsbergite CrN, a more abundant nitride in the Uakit meteorite, was not found in any assemblages with uakitite. Physical and optical properties of uakitite are quite similar to synthetic VN: yellow and transparent phase with metallic luster; Mohs hardness: 9–10; light gray color with a pinky tint in reflected light; density (calc.) = 6.128 g/cm$^3$. Uakitite is structurally related to the osbornite group minerals: carlsbergite CrN and osbornite TiN. Structural data were obtained for three uakitite crystals using the electron backscatter diffraction (EBSD) technique. Fitting of the EBSD patterns for a synthetic VN model (cubic, *Fm-3m*, *a* = 4.1328(3) Å; *V* = 70.588(9) Å$^3$; *Z* = 4) resulted in the parameter MAD = 0.14–0.37° (best-good fit). Analytical data for uakitite (*n* = 54, in wt. %) are: V, 71.33; Cr, 5.58; Fe, 1.56; N, 21.41; Ti, below detection limit (<0.005). The empirical formula $(V_{0.91}Cr_{0.07}Fe_{0.02})_{1.00}N_{1.00}$ indicates that chromium incorporates in the structure according to the scheme $V^{3+} \rightarrow Cr^{3+}$ (up to 7 mol. % of the carlsbergite end-member).

**Keywords:** uakitite; carlsbergite; osbornite group; troilite; daubréelite; Uakit meteorite; IIAB iron; Buryatia

## 1. Introduction

Nitrides and oxynitrides are very scarce minerals in natural conditions. Most of them (carlsbergite CrN, osbornite TiN, roaldite (Fe,Ni)$_4$N, nierite $\alpha$-Si$_3$N$_4$, sinoite Si$_2$N$_2$O) occur solely in extraterrestrial

environments, in different types of meteorites [1–20]. Only siderazot $Fe_5N_2$, which is "grandfathered" and questionable mineral (there is no modern confirmation for composition), appears to be terrestrial in origin, and was only identified in fumaroles associations of the Etna and Somma–Vesuvius volcanic complexes [21–23]. A new mineral, uakitite VN, was observed as an accessory phase in the Uakit iron meteorite and is the sixth nitride mineral which is found in meteorites [24–28]. The mineral was approved by the Commission on New Minerals, Nomenclature and Classification (CNMNC) of the International Mineralogical Association (IMA) as a new mineral species in May 2018 (IMA 2018-003) [26].

In contrast with the natural phase, synthetic VN is a well-known compound since the 1920s [29] and widely used in the different branches of industry. Like other transition metal nitrides (CrN, TiN, ZrN, NbN, etc.), it has long been of interest due to its excellent physical and chemical properties, such as high melting point, metallic conductivity, good chemical stability and high mechanical hardness. Owing to these properties, synthetic VN has a wide range of technological applications, e.g., as abrasive material, alloying component, wear and corrosion-resistant coatings, field emitter, supercapacitors, superconductors and buffer layers in microelectronics [30–45]. Intermediate compositions of TiN-CrN-VN have higher hardness than that for simple compounds [46,47].

## 2. History of the Uakit Meteorite

This iron meteorite was found summer 2016 by a gold prospector group during excavation works on river terrace (stream Mukhtunnyi, left feeder of the Uakit River) in 4 km west of the Uakit settlement, Baunt Evenk district, northern part of Republic of Buryatia, Russia (latitude: 55°29′47.50″ N; longitude: 113°33′47.98″ E). At present, one sample (3.96 kg, Figure 1) of the Uakit meteorite was identified. However, the information about the finding of larger iron meteorite mass ($\approx 50 \times 50$ cm) is known among prospectors around Uakit. The date of fall is unknown. During summer 2016, the 3.96 kg meteorite sample was passed to Oleg Yu. Korshunov (Ulan-Ude), who then handed several cut-offs in the Geological Institute (GI, SB RAS, Ulan-Ude, Russia) for expert examination. Later, some fragments of the meteorite were passed in V.S. Sobolev Institute of Geology and Mineralogy (IGM, SB RAS, Novosibirsk, Russia) for mineralogical studies in detail.

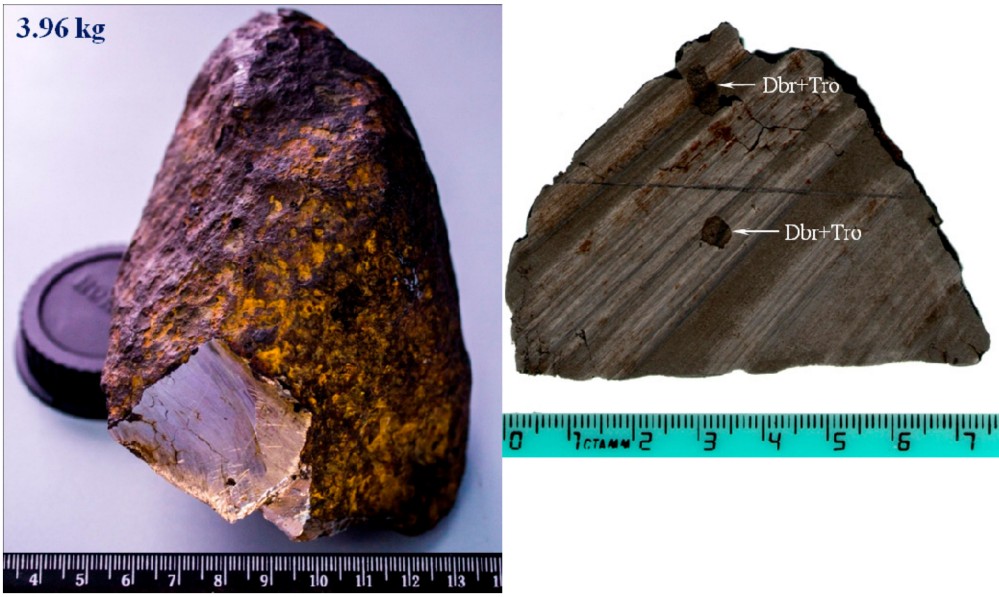

**Figure 1.** General view of the Uakit iron meteorite (IIAB) and one of cut-offs with troilite–daubréelite nodules. Symbols: Dbr + Tro—troilite–daubréelite nodule.

The Uakit iron meteorite (IIAB) was approved on the 28 June 2017 by the Meteorite Nomenclature Committee (see Meteoritical Bulletin Database). The cut-off fragments of the meteorite are now

deposited in the meteorite collections of the Central Siberian Geological Museum of the IGM (CSGM IGM, type specimens: 70.3 g and 17.5 g cut-offs, SB RAS, Novosibirsk, Russia), the Museum of the Buryatian Scientific Centre (51.2 and 18 g cut-offs, MBSC SB RAS, Ulan-Ude, Russia), ExtraTerra Consortium Lab, Ural Federal University (1.6 g cut-off, Ekaterinburg, Russia) and A.E. Fersman Mineralogical Museum (<1 g cut-off, Moscow, Russia).

As a result of detailed mineralogical studies two new minerals, were identified in this meteorite: uakitite VN (IMA 2018-003) [26] and grokhovskyite $CuCrS_2$ (IMA 2019-065) [48]. In this paper, we provide a detailed description of uakitite. Some data on this mineral in the Uakit meteorite were reported in a few previous publications [24–26,48,49]. The name of the mineral is given after the meteorite. The holotype specimens of uakitite are on display in the meteorite collections of the CSGM IGM, SB RAS, Novosibirsk (registration numbers 52 and 52b, meteorite Uakit) and in the MBSC, SB RAS, Ulan-Ude (registration number Uakit-MBSC435/G84).

## 3. Analytical Methods

Polished fragments were used for optical examination of the Uakit meteorite in reflected light. The identification of all minerals was based on energy-dispersive spectra (EDS), backscattered electron (BSE) images and elemental mapping (EDS system), using a TESCAN MIRA 3MLU scanning electron microscope equipped with an INCA Energy 450 XMax 80 microanalysis system (Oxford Instruments Ltd., Abingdon, UK) at the IGM, Novosibirsk, Russia, and a LEO-1430 scanning electron microscope equipped with an INCA Energy-300 EDS microanalysis system at the GI, Ulan-Ude, Russia. The instruments were operated at an accelerating voltage of 20 kV and a probe current of 1 nA in high-vacuum mode. EDS analyses of uakitite and other minerals were done at an accumulation time of 20–40 s. The following simple compounds and metals were used as reference standards for most of the elements: $Ca_2P_2O_7$ (P), $Cr_2O_3$ (Cr), pyrite (S), $Si_3N_4$ or BN (N), metallic Ti, Fe, Cu, Zn, Mn, Ni, V and others. Correction for matrix effects was done using the XPP algorithm, implemented in the software of the microanalysis system. Metallic Co served for quantitative optimization (normalization to probe current and energy calibration of the spectrometer). The overlapping of $VK\beta$ and $CrK\alpha$ was specially checked using the $Cr_2O_3$ and metallic V standards.

Electron microprobe analyses (EMPA) in wavelength-dispersive (WDS) mode were performed for metals and sulfides, which are associated with uakitite in the Uakit iron meteorite, using a JXA-8100 microprobe (Jeol Ltd., Tokyo, Japan) at IGM. Grains (sizes > 5 µm) previously analyzed by EDS were selected for this purpose. The operating conditions were as follows: beam diameter of 1–2 µm, accelerating voltage of 20 kV, beam current of 50 nA and counting time of 10 (5 + 5) s. The following standards were used for two microprobe sessions: natural spessartite (Mn), synthetic FeS or $FeS_2$ (S), Fe-metal or FeS (Fe), Fe-Ni-Co alloy (Ni and Co), GaP (P), $Cr_2O_3$ (Cr), ZnS (Zn), $CuFeS_2$ (Cu) and $V_2O_5$ (V). Correction for matrix effects was done using a PAP routine [50]. The precision of analysis for major elements was better than 2% relative. The detection limits for elements were (in ppm): S, 115–208; Fe, 132–188; Ni, 119–130; Co, 55–102; P, 167–185; Mn, 184–203; Cr, 146–170; Zn, 108–119; Cu, 108; V, 164.

Electron backscatter diffraction (EBSD) studies were provided for three grains of uakitite. Samples containing uakitite and intended for EBSD studies were subjected to polishing by BuehlerMasterMet2 non-crystallizing colloidal silica suspension (0.02 µm). EBSD measurements were carried out by means of an FE-SEM ZEISS SIGMA VP scanning electron microscope equipped with an Oxford Instruments Nordlys HKL EBSD detector, operated at 20 kV and 1.4 nA in focused beam mode with a 70° tilted stage at Institute of Physics and Technology, Ural Federal University, Ekaterinburg, Russia. Structural identification of uakitite was performed by matching its EBSD patterns with the reference structural models using program FLAMENCO.

## 4. General Description of the Uakit Meteorite

The 3.96 kg mass of the meteorite is oval (10 × 10 × 7 cm). The exterior part is covered by thin crust of brown to yellow-brown secondary products (mainly, different Fe-rich hydroxides, Figure 1). Polished and then etched surfaces of the meteorite cut-offs show the presence of large (≈ 2 cm) kamacite crystals with evident Neumann lines; no Widmannstatten pattern is observed (Figure 2). Weathering and fusion crusts are less than 1 mm in the exterior. Sometimes they extend together along some fractures of the outer part, but their abundance is low (Figure 1). The shock stage is medium and mainly fixed by shifting of blocks in some schreibersite and carlsbergite crystals and by Neumann lines.

The bulk composition of the meteorite is (ICP-MS, IGM, $n$ = 2): Ni = 5.47; Co = 0.45 (both in wt. %); Si = 732–886; P = 989–1063; Cr = 127–139; Cu = 149–294; V = 0.10–0.24; Zn = 6–68; Ga = 49–50; Ge = 203–215; As = 2.4–3.0; Mo = 5.1–5.9; Ru = 18.7; Rh = 2.0; Pd = 1.26–1.40; Sn = 6.9–44; Sb = 0.064–0.10; W = 3.0; Re = 1.67–1.76; Ir = 20; Pt = 24; Au = 0.51–0.53 (in ppm) [25]. The Uakit iron meteorite is structurally and geochemically characterized to be a hexahedrite, IIAB group, with tendency to the IIA subgroup.

Fe-Ni-metal (kamacite) is the main mineral of the meteorite (≈ 93–98 vol. %, Figure 1). Minor and accessory primary minerals are represented by schreibersite (rhabdite), nickelphosphide, taenite, plessite (taenite + kamacite + tetrataenite), cohenite, tetrataenite, daubréelite, kalininite, troilite, carlsbergite, sphalerite, uakitite, copper, grokhovskyite and an unidentified Mo-dominant phase (<0.5 µm, molybdenite or hexamolybdenum (Mo,Ru,Fe) or Mo or MoC, according to recent meteorite minerals list in [19]) (Figures 1–3).

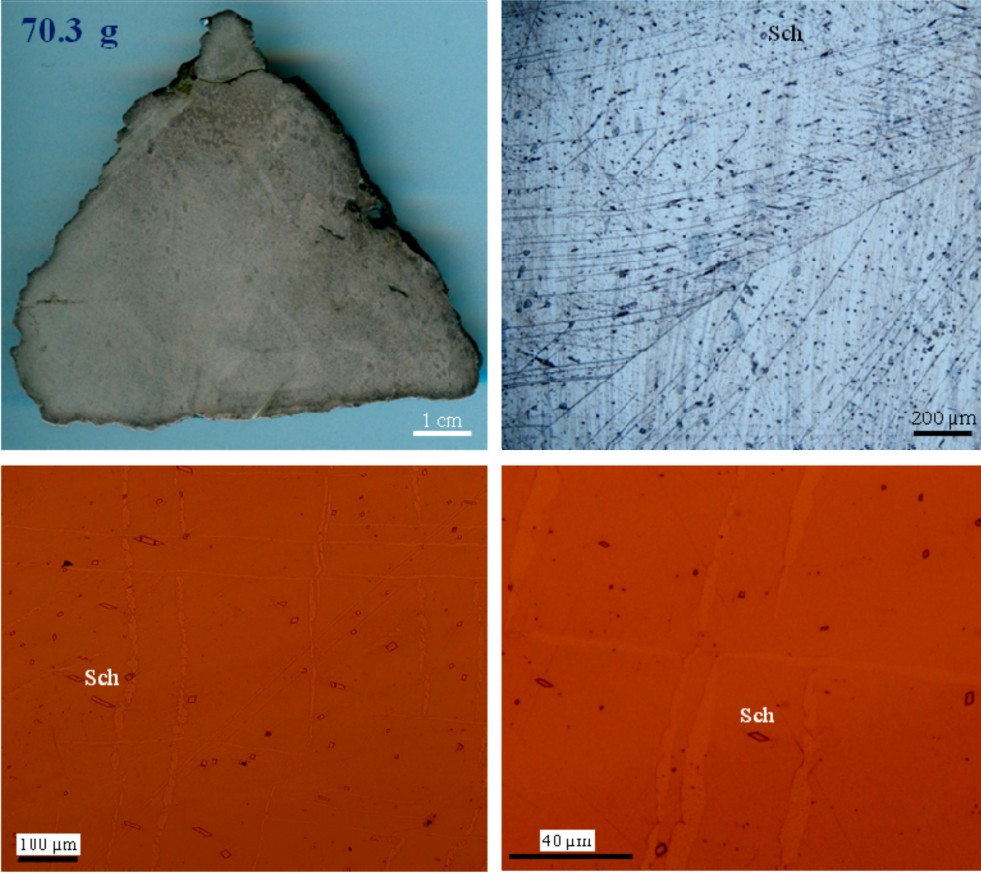

**Figure 2.** Neumann lines and oriented schreibersite grains (Sch) in Fe-Ni-metal (kamacite), Uakit meteorite (IIAB), images in ordinary and reflected light. Upper images: samples were etched by nital. The upper left image is for type specimen (70.3 g cut-off) of the Uakit meteorite from the CSGM IGM, SB RAS, Novosibirsk.

The appearance of Ni-rich magnetite, pentlandite, heazlewoodite, awaruite to native nickel, unidentified Ni-Fe-Cr-sulfide as well as Ni-rich goethite, akaganeite, Ni-rich siderite, Ca-Fe-carbonates, gypsum and unidentified hydrated Fe-rich phosphate and Ca-Fe-sulfate is related to different stages of the terrestrial alteration [24–26,48,49]. Magnetite, pentlandite, awaruite-nickel and heazlewoodite seem to be related to the high-temperature alteration (fusion crust?), whereas goethite and other Fe-hydroxides to low-temperature weathering products. The chemical composition of the principal minerals in the Uakit iron meteorite is listed in Table 1.

The presence of large sulfide nodules (up to 1 cm) of troilite–daubréelite composition is common in the Uakit meteorite (Figure 1). Cohenite occurs mainly near the exterior and forms skeletal crystals up to 0.5 mm; sometimes it is observed on the boundary between kamacite grains. Ni-poorer taenite (<33 wt. % Ni), Ni-rich kamacite (>6.3 wt. % Ni) and Fe-rich nickelphosphide are most typical for the cohenite-containing associations.

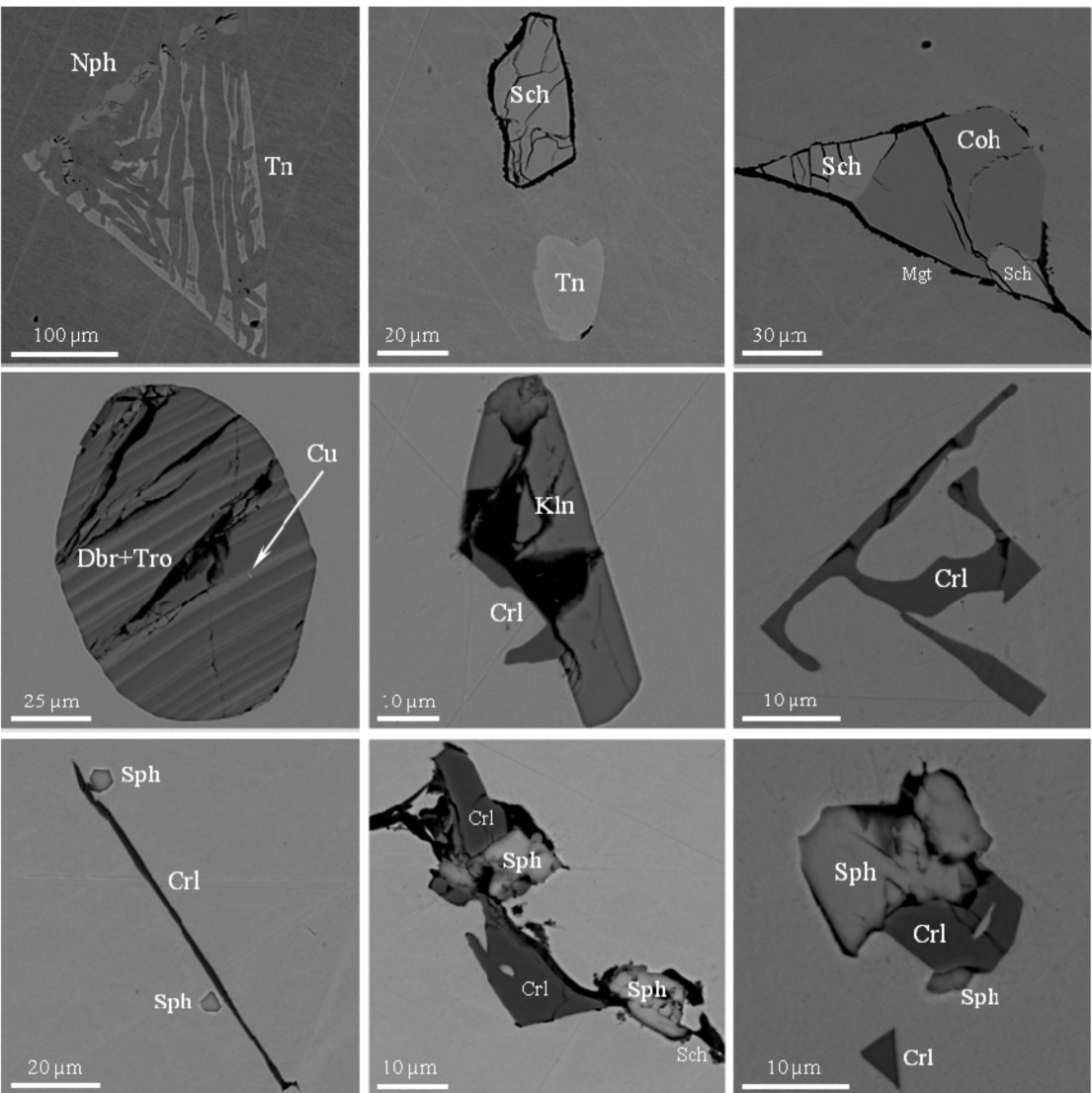

**Figure 3.** Mono- and polymineralic inclusions and globules in Fe-Ni-metal (kamacite), Uakit meteorite (IIAB), BSE images. Symbols: Dbr—daubréelite; Sch—schreibersite; Nph—nickelphosphide; Tn—taenite; Coh—cohenite; Tro—troilite; Cu—native copper; Crl—carlsbergite; Kln—kalininite; Sph—sphalerite; Mgt—magnetite.

**Table 1.** Chemical composition (WDS, wt. %) of the essential minerals in the Uakit iron meteorite.

| Mineral | | Fe | Mn | Ni | Co | Zn | Cu | Cr | V | P | S | Sum | Formula based on | Fe | Mn | Ni | Co | Zn | Cu | Cr | V | P | S |
|---|---|---|---|---|---|---|---|---|---|---|---|---|---|---|---|---|---|---|---|---|---|---|---|
| Kamacite | n = 34 | 93.46 | n.d. | 6.09 | 0.48 | n.d. | n.d. | n.d. | n.d. | 0.06 | n.d. | 100.09 | 1 ion | 0.94 | | 0.06 | 0.00 | | | | | 0.00 | |
| | sd | 0.56 | | 0.51 | 0.05 | | | | | 0.04 | | | | | | | | | | | | | |
| Taenite | n = 19 | 67.63 | n.d. | 32.12 | 0.13 | n.d. | 0.23 | n.d. | n.d. | n.d. | n.d. | 100.10 | 1 ion | 0.69 | | 0.31 | 0.00 | | 0.00 | | | | |
| | sd | 6.48 | | 6.54 | 0.05 | | 0.09 | | | | | | | | | | | | | | | | |
| Tetrataenite | n = 14 | 42.66 | n.d. | 56.96 | 0.01 | n.d. | 0.24 | 0.17 | n.d. | n.d. | n.d. | 100.04 | 1 ion | 0.44 | | 0.56 | 0.00 | | 0.00 | 0.00 | | | |
| | sd | 1.08 | | 1.22 | 0.02 | | 0.10 | 0.15 | | | | | | | | | | | | | | | |
| Cohenite | n = 18 | 91.64 | n.d. | 1.57 | 0.08 | n.d. | n.d. | 0.12 | n.d. | n.d. | n.d. | 93.41 | 3 ions | 2.95 | | 0.05 | 0.00 | | | 0.00 | | | |
| | sd | 0.29 | | 0.06 | 0.02 | | | 0.05 | | | | | | | | | | | | | | | |
| Schreibersite | n = 19 | 50.47 | n.d. | 34.25 | 0.09 | n.d. | n.d. | 0.04 | n.d. | 15.16 | 0.01 | 100.03 | 4 ions | 1.83 | | 1.18 | 0.00 | | | 0.00 | | 0.99 | 0.00 |
| | sd | 3.56 | | 3.51 | 0.02 | | | 0.09 | | 0.08 | 0.02 | | | | | | | | | | | 0.99 | 0.00 |
| Nickelphosphide | n = 12 | 39.43 | n.d. | 45.02 | 0.05 | n.d. | n.d. | 0.25 | n.d. | 15.17 | 0.05 | 99.97 | 4 ions | 1.43 | | 1.56 | 0.00 | | | 0.01 | | 0.99 | 0.00 |
| | sd | 2.81 | | 2.74 | 0.04 | | | 0.37 | | 0.06 | 0.04 | | | | | | | | | | | | |
| Daubréelite | n = 39 | 19.22 | 0.23 | 0.07 | 0.01 | 0.48 | 0.07 | 35.47 | 0.01 | n.d. | 44.42 | 99.91 | 7 ions | 0.99 | 0.01 | 0.00 | 0.00 | 0.02 | 0.00 | 1.97 | 0.00 | | 4.00 |
| | sd | 0.84 | 0.20 | 0.09 | 0.02 | 0.53 | 0.03 | 0.63 | 0.02 | | 0.11 | | | | | | | | | | | | |
| Kalininite | n = 11 | 10.15 | 0.04 | n.d. | n.d. | 11.50 | n.d. | 34.55 | n.d. | n.d. | 43.68 | 99.92 | 7 ions | 0.53 | 0.00 | | | 0.52 | | 1.95 | | | 4.00 |
| | sd | 0.92 | 0.00 | | | 0.91 | | 0.20 | | | 0.05 | | | | | | | | | | | | |
| Troilite | n = 13 | 62.42 | n.d. | 0.16 | n.d. | n.d. | 0.12 | 0.67 | 0.01 | n.d. | 36.53 | 99.92 | 2 ions | 0.98 | | 0.00 | | | 0.00 | 0.01 | | | 1.00 |
| | sd | 0.32 | | 0.29 | | | 0.03 | 0.18 | 0.02 | | 0.07 | | | | | | | | | | | | |
| Pentlandite | n = 9 | 32.87 | n.d. | 31.75 | 1.39 | n.d. | 0.01 | 0.68 | 0.02 | n.d. | 33.27 | 99.98 | 17 ions | 4.54 | | 4.17 | 0.18 | | 0.00 | 0.10 | 0.00 | | 8.00 |
| | sd | 3.48 | | 2.79 | 1.13 | | 0.02 | 0.16 | 0.02 | | 0.05 | | | | | | | | | | | | |
| Heazlewoodite | n = 5 | 5.68 | n.d. | 64.77 | 1.57 | n.d. | n.d. | 0.45 | 0.02 | n.d. | 27.45 | 99.94 | 5 ions | 0.24 | | 2.63 | 0.06 | | | 0.02 | 0.00 | | 2.04 |
| | sd | 0.70 | | 2.20 | 1.27 | | | 0.29 | 0.02 | | 1.02 | | | | | | | | | | | | |

*n*—number of analyses; sd—standard deviation; n.d.—not detected.

Schreibersite (20–100 μm) and carlsbergite (1–10 μm) elongated crystals are sometimes oriented in one or more directions within large kamacite grains (Figure 2). In addition to the above assemblages, the presence of rounded sulfide globules (mainly daubréelite + troilite, up to 100 μm), plessite isolations (taenite + kamacite + tetrataenite) and mono- or polymineralic inclusions are very common in kamacite (Figures 3–7). Sulfide globules with a "layered structure" (alternating layers of troilite and daubréelite as a possible result of solid decay of initial high-temperature Fe-Cr-sulfide) are more widespread than those without layering. Namely, in such associations two new minerals (uakitite, grokhovskyite), Fe-rich kalininite (first finding in meteorites), copper and sphalerite were identified in this meteorite (Figure 3).

## 5. Morphology, Optical and Physical Properties of Uakitite

At present, uakitite is observed only in small troilite–daubréelite (±schreibersite) globules with "layered structure" hosted by kamacite and in large troilite–daubréelite nodules (Figures 1 and 4–8). It forms isometric (cubic) crystals (in daubréelite) or rounded grains (in schreibersite). The size of uakitite grains is commonly less than 5 μm (Figures 4–8); the largest detected crystal is 5 × 5 μm (Figures 4 and 5). Twinning was not observed. Uakitite from sulfide globules is associated with sulfides (daubréelite, troilite, grokhovskyite), schreibersite and magnetite. In large sulfide nodules, it is confined to their outer margins (Figure 7). In general, both globules and nodules exhibit varying degrees of partial resorption due to alteration processes. Carlsbergite CrN is a more abundant nitride in the Uakit meteorite, but it was not found yet in any assemblages with uakitite. In addition to polymineralic inclusions (Figure 3), carlsbergite also occurs in troilite–daubréelite globules without "layered structure", where its micron-sized crystals may decorate the boundary between globule and host kamacite [49]. The phase relationships in the sulfide associations indicate that uakitite is one of the early minerals (Figures 5 and 6) and crystallized under temperature higher than that for troilite–daubréelite solid decay (>1000 °C).

We were unable to obtain physical and optical properties of uakitite due to the very small sizes of the grains. So, in most cases, we have to refer to data for synthetic VN. It has a yellow color, white streak and metallic luster. The mineral is transparent, non-fluorescent and brittle. No cleavage and parting are observed. The hardness for synthetic VN is ≈ 9–10 (Mohs), microhardness: VHN load: 0.5–0.98 mN; range: 6.0–11.8 GPa [33,46,47]. The density (6.128 g/cm$^3$) for uakitite was calculated from unit-cell dimensions and results of EDS analyses. Under reflected light, uakitite is light gray with a pinky tint and does not show any internal reflections. Optical property data (transmitted light) for synthetic VN (at 0.5876 μm) are: refractive index $n$ = 2.3031, $N_g$ = 1.4501, reflectance $R$ = 0.43817 [31]. Uakitite is not soluble in $H_2O$ and weakly concentrated in HCl, $HNO_3$ and $H_2SO_4$. On BSE images, it resembles magnetite and daubréelite.

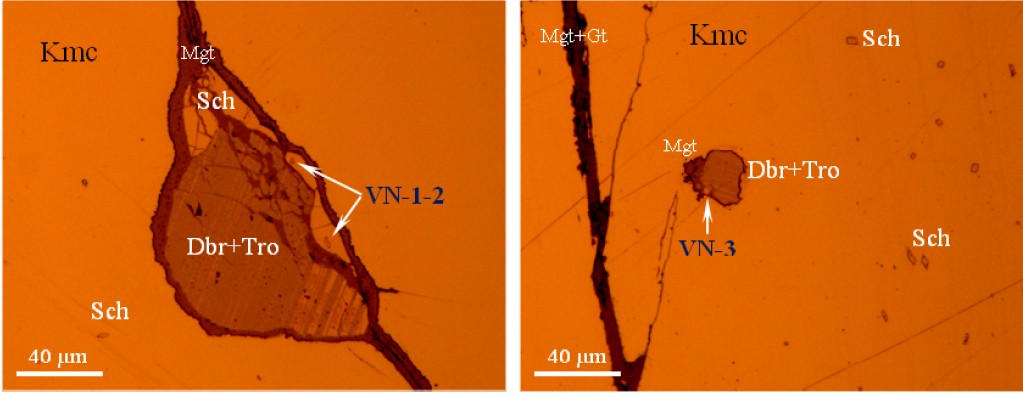

**Figure 4.** Uakitite in troilite+daubréelite±schreibersite globules in Fe-Ni-metal (kamacite), Uakit meteorite (IIAB), images in reflected light. Symbols: VN-1–VN-3—uakitite; Dbr—daubréelite; Tro—troilite; Sch—schreibersite; Mgt—magnetite; Gt—goethite; Kmc—kamacite.

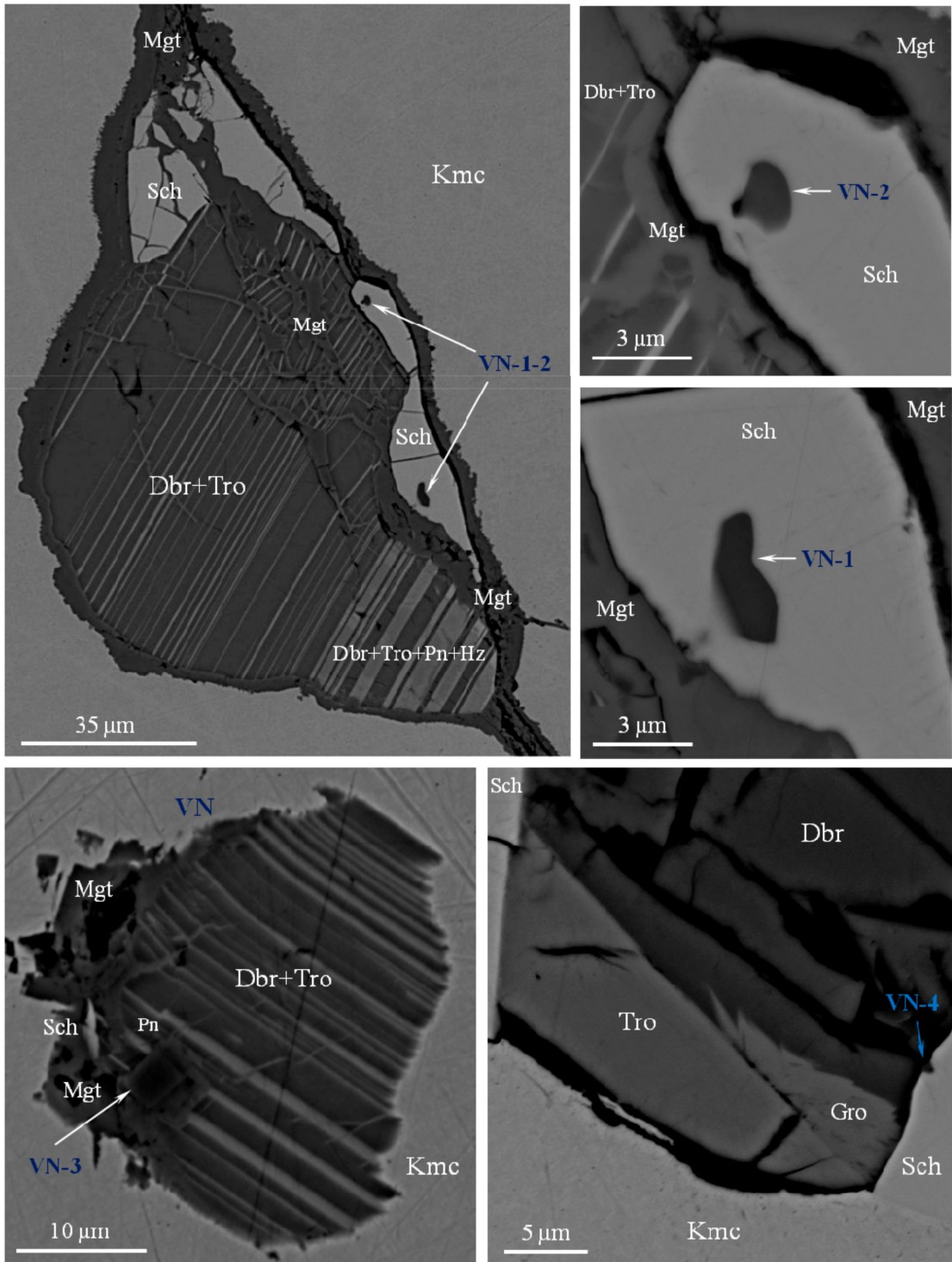

**Figure 5.** Uakitite in troilite–daubréelite ± schreibersite globules in Fe-Ni-metal (kamacite), Uakit meteorite (IIAB), BSE images. Symbols: VN-1–VN-4—uakitite (see also Figure 4); Dbr—daubréelite; Tro—troilite; Sch—schreibersite; Mgt—magnetite; Kmc—kamacite, Pn—pentlandite; Hz—heazlewoodite; Gro—grokhovkyite $CuCrS_2$.

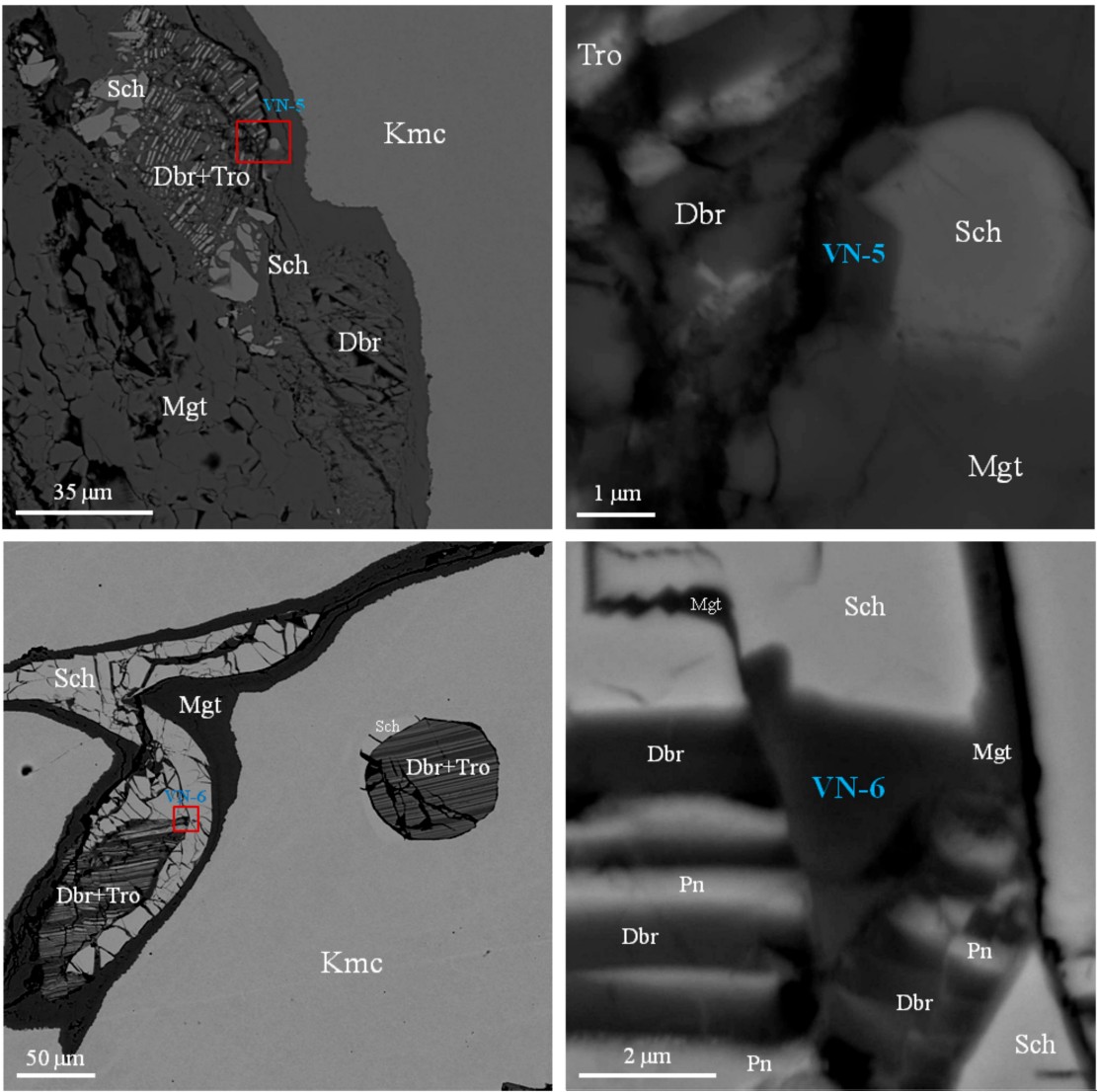

**Figure 6.** Uakitite in resorbed troilite–daubréelite–schreibersite globules from fissures filled with magnetite, Uakit meteorite (IIAB), BSE images. Symbols: VN-5, VN-6—uakitite; Dbr—daubréelite; Tro—troilite; Sch—schreibersite; Mgt—magnetite; Kmc—kamacite.

## 6. Chemical Composition of Uakitite

The presence of elevated Cr and Fe and absence of Ti are characteristic features of uakitite. Its empirical formula ($n = 54$) based on two ions is $V_{0.91}Cr_{0.07}Fe_{0.02}N_{1.00}$ (Table 2). In general, the variations in all components are negligible for individual uakitite grains. It is supported by elemental maps for some grains (Figure 8). The ideal formula for uakitite is VN, which requires V 78.43, N 21.57 and total 100.00 wt. % (Table 2). Uakitite is structurally related to carlsbergite CrN and osbornite TiN [1,3,7–10,18]. The essential impurity of chromium incorporates in uakitite according to the scheme $V^{3+} \rightarrow Cr^{3+}$, up to 7 mol. % of the carlsbergite end-member (Table 2, Figure 9). The isomorphic scheme for insignificant Fe (1.2–2.1 wt. %) is unclear. Two variants are possible: $V^{3+} \rightarrow Fe^{3+}$ and $2V^{3+} \rightarrow V^{4+} + Fe^{2+}$ (Figure 9). In contrast to uakitite, the concentration of V in carlsbergite is less than 0.2 wt. %; Fe is up to 4.5 wt. %; Ti was not detected (Table 2, Figure 9). In general, conditions for accumulation of V as VN are not yet clear. Bulk compositions of the Uakit meteorite and kamacite (ICP-MS and LA-ICP-MS) indicate very low vanadium concentrations of 0.04–0.52 ppm [25].

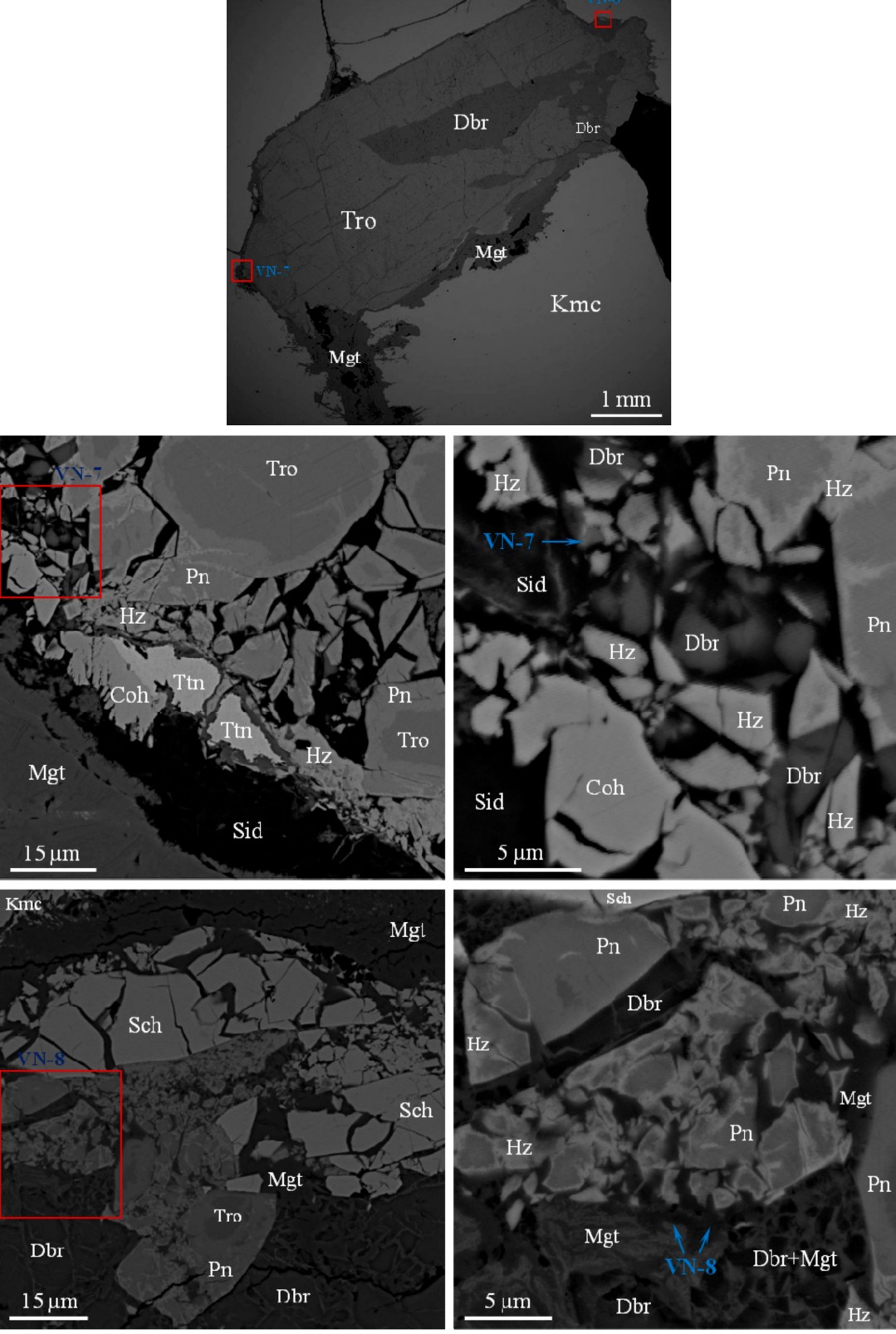

**Figure 7.** Uakitite in a large troilite–daubréelite nodule, Uakit meteorite (IIAB), BSE images. Symbols: VN-7, VN-8—uakitite; Dbr—daubréelite; Tro—troilite; Sch—schreibersite; Mgt—magnetite; Kmc—kamacite; Pn—pentlandite; Hz—heazlewoodite; Coh—cohenite; Ttn—tetrataenite; Sid—siderite.

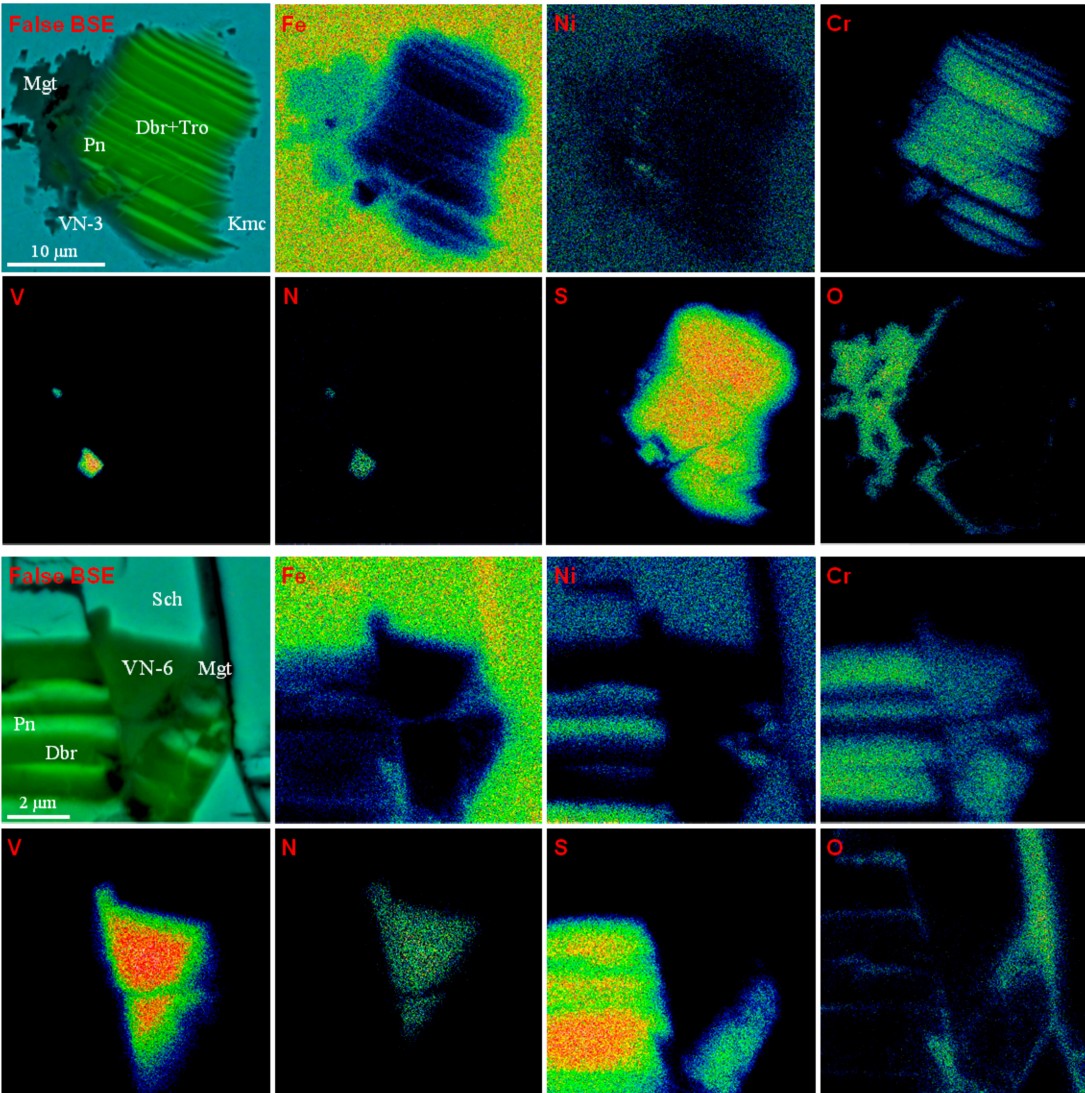

**Figure 8.** Elemental maps for troilite–daubréelite globules with uakitite. Symbols: VN-3, VN-6—uakitite (see Figures 5 and 6); Mgt—magnetite; Dbr+Tro—daubréelite+troilite; Kmc—kamacite; Sch—schreibersite; Pn—pentlandite.

**Table 2.** Chemical composition (EDS, wt. %) of uakitite in comparison with ideal compositions and carlsbergite from the Uakit meteorite.

| | VN (All Grains) | | | | VN-1 | | VN-2 | | VN-3 | | VN-6 | Ideal-1 | Ideal-2 | CrN | |
|---|---|---|---|---|---|---|---|---|---|---|---|---|---|---|---|
| | Mean | sd | Min | Max | Mean | sd | Mean | sd | Mean | sd | Mean | VN | $V_{0.9}Cr_{0.1}N$ | Mean | sd |
| | *n* = 54 | | | | *n* = 34 | | *n* = 9 | | *n* = 10 | | *n* = 1 | | | *n* = 47 | |
| V | 71.33 | 0.21 | 70.91 | 71.90 | 71.33 | 0.22 | 71.24 | 0.13 | 71.42 | 0.24 | 71.07 | 78.43 | 70.48 | 0.06 | 0.08 |
| Cr | 5.57 | 0.27 | 5.02 | 6.18 | 5.59 | 0.31 | 5.57 | 0.19 | 5.53 | 0.18 | 5.43 | 0.00 | 7.99 | 76.64 | 0.73 |
| Fe | 1.56 | 0.22 | 1.16 | 2.08 | 1.54 | 0.24 | 1.66 | 0.17 | 1.54 | 0.21 | 2.08 | 0.00 | 0.00 | 2.18 | 0.73 |
| N | 21.41 | 0.07 | 21.22 | 21.54 | 21.41 | 0.07 | 21.40 | 0.07 | 21.44 | 0.05 | 21.38 | 21.57 | 21.53 | 21.13 | 0.08 |
| Sum | 99.88 | | | | 99.87 | | 99.86 | | 99.93 | | 99.96 | 100.00 | 100.00 | 100.01 | |
| *Formula based on 2 ions* | | | | | | | | | | | | | | | |
| V | 0.914 | | | | 0.914 | | 0.913 | | 0.914 | | 0.911 | 1.000 | 0.900 | 0.001 | |
| Cr | 0.070 | | | | 0.070 | | 0.070 | | 0.069 | | 0.068 | 0.000 | 0.100 | 0.975 | |
| Fe | 0.018 | | | | 0.018 | | 0.019 | | 0.018 | | 0.024 | 0.000 | 0.000 | 0.026 | |
| N | 0.998 | | | | 0.998 | | 0.998 | | 0.998 | | 0.997 | 1.000 | 1.000 | 0.998 | |

VN-1–VN-8—individual uakitite grains (see Figures 5, 6 and 8); CrN—carlsbergite. Ti is below detection limit (<0.005 wt. %). sd—standard deviation; Ideal-1—VN and Ideal-2—$V_{0.9}Cr_{0.1}N$.

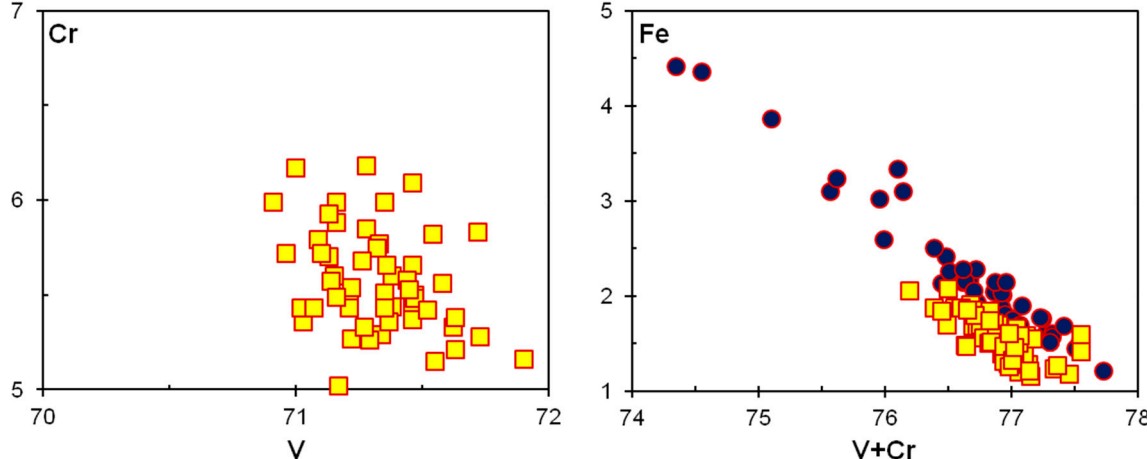

**Figure 9.** Chemical variations (in wt. %) for uakitite and carlsbergite from the Uakit meteorite. Symbols: squares—uakitite; circles—carlsbergite.

## 7. Crystal Structural Data for Uakitite

It was difficult to obtain single-crystal and X-ray powder diffraction data for uakitite because of its very small crystal size (<5 μm), and its mineral structure was resolved by EBSD method. Before considering the structure of uakitite, it is important to consider the data for the V-N system and the structure of the synthetic VN phase.

### 7.1. System V-N

Three stable solid phases are known in the system V-N: V, $V_2N$ and VN [51–61]. Cubic $VN_{1-x}$ and hexagonal $V_2N_{1-x}$ are dominant solids in a very broad temperature range (Figure 10) according to [59,60]. Compound VN (or $VN_{1-x}$ or δ-VN) was firstly synthesized in the 1920s [29]. It is cubic (NaCl-type structure, *Fm-3m*, $a \approx 4.135$ Å, Z = 4): at high temperature (near melting point, 2050–2119 °C), its composition is shifted to $VN_{0.7-0.8}$, whereas at room temperature it is close to stoichiometric VN. At temperatures below 230 K, it transforms into a tetragonal, noncentrosymmetric low-temperature modification (*P-42m*, $a = 4.1314(3)$ Å, $c = 4.1198(3)$ Å at 45 K) [62–64]. The melting point of cubic VN is increased up to 2800 °C at pressure 10 MPa [58].

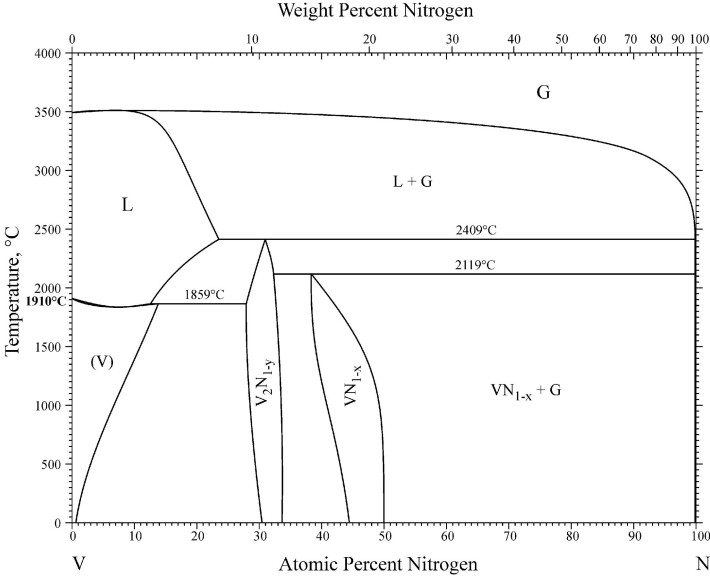

**Figure 10.** The phase diagram V-N at 1 atm [59,60].

### 7.2. Crystal Structure for Synthetic VN

In general, the crystal structure and properties of well-known synthetic VN (or δ-VN) have been studied in detail [29,30,40,42,44,51,62,64–82]. It is a cubic NaCl-type structure (*Fm-3m*, $a \approx 4.135$ Å, $Z = 4$). The δ-VN is isostructural with other transition metal nitrides (CrN, TiN, ZrN, NbN, etc.). The crystal structure of synthetic VN is shown in Figure 11.

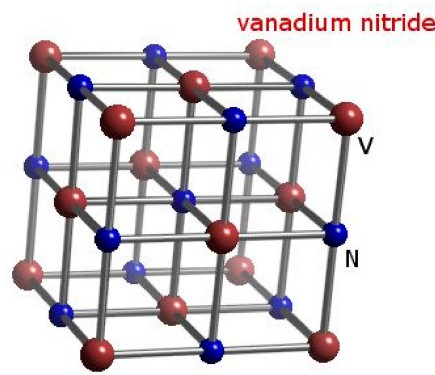

**Figure 11.** The crystal structure of synthetic VN [62].

### 7.3. EBSD Data for Uakitite

Single-crystal X-ray studies could not be carried out because of the small crystal size of uakitite. Structural data were obtained using the EBSD technique (Figure 12) and fitted to the following structural model of synthetic VN (space group *Fm-3m* (225); $a = 4.1328(3)$ Å; $V = 70.588(9)$ Å$^3$; $Z = 4$) [62]. The EBSD patterns for three uakitite crystals were obtained at working distances of 15–20 mm. Fitting of the EBSD patterns for a VN model with the cell parameters given below resulted in the parameter MAD = 0.14–0.37° (best-good fit). EBSD studies showed full structural identity between uakitite and its synthetic analog VN (NaCl-type). Uakitite is structurally related to the osbornite group which also includes carlsbergite CrN and osbornite TiN [1,3,7–10,18].

### 7.4. Diffraction Data for Uakitite

Because uakitite occurs only in small concentrations, X-ray powder diffraction data were not collected. The theoretical powder diffraction pattern was calculated using the structural data of the synthetic analog [62] and the empirical formula of uakitite (Table 2). Data are given in Table 3. Calculated structure data for uakitite are presented in Supplementary Materials (Cif file).

**Table 3.** Calculated powder diffraction data for uakitite.

| *h* | *k* | *l* | $d_{calc}$, Å | $I_{rel}$ |
|---|---|---|---|---|
| **1** | **1** | **1** | **2.386** | **71.22** |
| **2** | **0** | **0** | **2.066** | **100.00** |
| **2** | **2** | **0** | **1.461** | **61.15** |
| **3** | **1** | **1** | **1.246** | **29.12** |
| **2** | **2** | **2** | **1.193** | **18.92** |
| 4 | 0 | 0 | 1.033 | 8.03 |
| **3** | **3** | **1** | **0.948** | **10.16** |
| **4** | **2** | **0** | **0.924** | **20.55** |
| **4** | **2** | **2** | **0.844** | **14.29** |
| 5 | 1 | 1 | 0.795 | 4.99 |
| 3 | 3 | 3 | 0.795 | 1.66 |
| 4 | 4 | 0 | 0.731 | 4.08 |

MoK$\alpha$1 = 0.70932 Å, Bregg–Brentano geometry, fixed slit, no anomalous dispersion, $I > 1$; data were calculated using PowderCell 2.4 [83]. The strongest diffraction lines are given in bold.

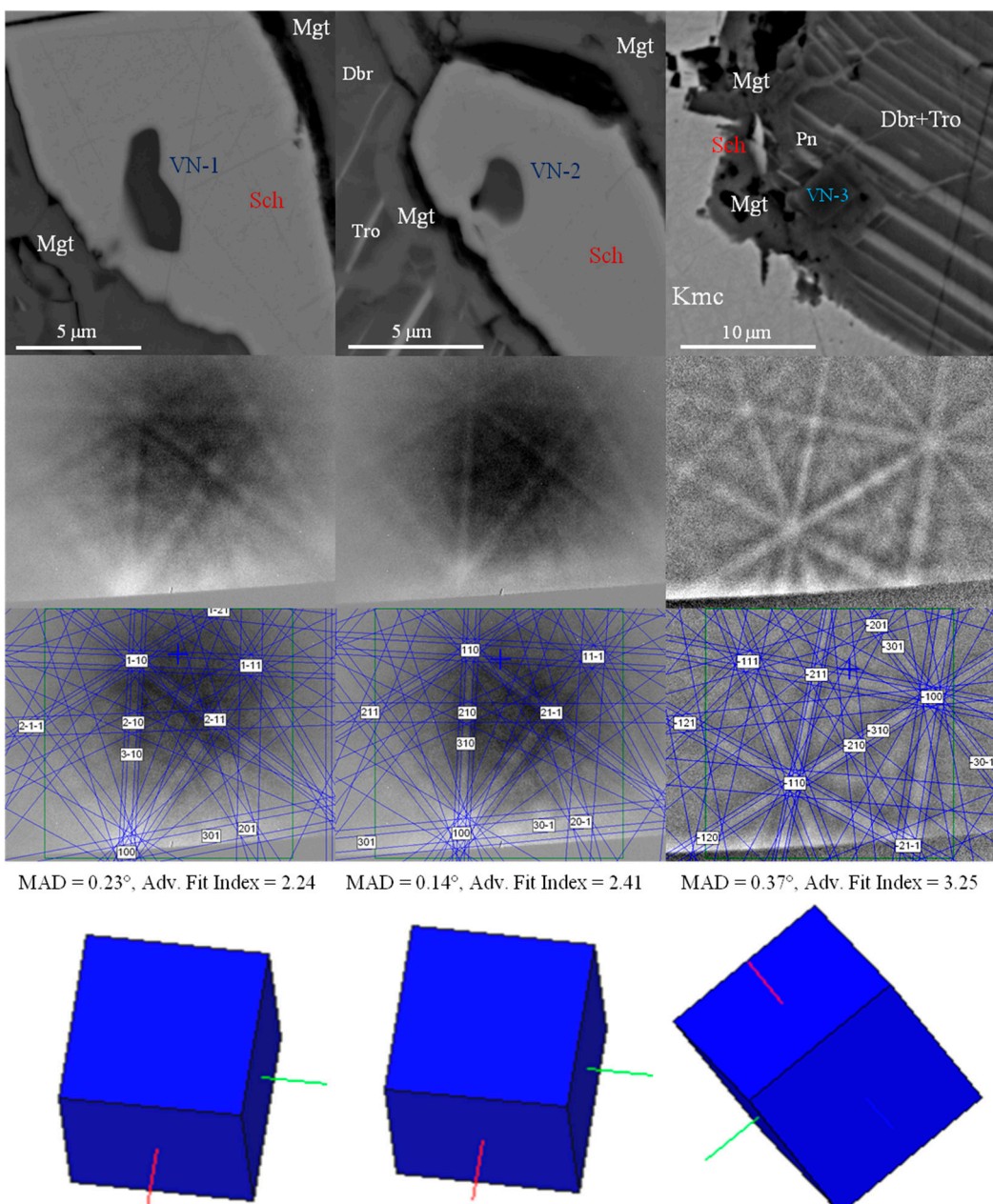

**Figure 12.** Electron backscattered diffraction (EBSD) patterns, the Kikuchi patterns and orientation for three grains of uakitite (detector distance: 15–20 mm). Symbols: VN-1–VN-3—uakitite (see Figure 5); Mgt—magnetite; Sch—schreibersite; Pn—pentlandite; Dbr + Tro—daubréelite + troilite; MAD—mean angular deviation.

## 8. Discussion and Concluding Remarks

The detailed mineralogical and petrographic studies for the Uakit iron meteorite gave a possibility to describe the chemical composition and some structural affinities for a new mineral, uakitite VN, which belongs to the osbornite group. Phase relations indicate that uakitite is one of the early minerals in the troilite–daubréelite associations. These sulfide associations in the Uakit meteorite seemed form due to high-temperature (>1000 °C) separation of Fe-Cr-rich sulfide liquid from Fe-metal melt. We do not exclude that crystallization of uakitite was under high temperature (≈ 1000 °C) from the sulfide melt, but was not below 650 ± 50 °C according to the system Cr-Fe-S [84]. In general, conditions for high accumulation of V as VN are not yet clear. Bulk compositions of whole meteorite and

kamacite (ICP-MS and LA-ICP-MS) indicate very low vanadium concentrations 0.04–0.52 ppm [25]. Probably sulfide-metal liquid immiscibility is the main factor for the partitioning of chalcophile V (and also Cr) in sulfide melt.

The discovery of uakitite in conjunction with occurrences for other extraterrestrial nitrides indicates the very interesting regularity in their appearance [1–20]. Carlsbergite, uakitite and roaldite are characteristic minerals of iron meteorites, whereas osbornite, nierite and sinoite occur solely in stone meteorites (carbonaceous chondrites, enstatite chondrites and achondrites). Possibly, it is also related to the chalcophile character of elements.

Some rare and exotic minerals occur as very minute grains (size: <1–20 μm and smaller). It creates a lot of problems in their identification and detailed description; especially in regards to new mineral species (composition, unit-cell data and crystal structure). However, modern analytical methods permit the study of such small objects. In addition to the classic analytical methods, the application of the TEM, EBSD and other techniques allow for improved studies of micron-sized minerals. In the last decades, these technologies are successfully used for detailed identification of new minerals in both meteorites and terrestrial rocks, especially when their synthetic analogs are known (for example [26,48,85–89] and many other works).

**Supplementary Materials:** The following are available online at http://www.mdpi.com/2075-163X/10/2/150/s1, Cif file: uakitite.

**Author Contributions:** V.V.S. and G.S.R. wrote the paper. V.V.S. performed the mineralogical description and measurements of chemical composition of uakitite and related minerals (EMPA). G.S.R., I.A.I. and E.A.K. provided preliminary SEM studies. G.A.Y., V.I.G., E.A.K., N.S.K., I.A.I. and Y.V.S. provided EBSD, SEM and structural studies. All authors have read and agreed to the published version of the manuscript.

**Funding:** The investigations were partly supported by RFBR (grant 17-05-00129) and the State assignment project (IGM SD 0330-2016-0005). This work was also supported by the Initiative Project of Ministry of Science and Higher Education of the Russian Federation and by Act 211 of the Government of the Russian Federation, agreement no. 02.A03.21.0006.

**Acknowledgments:** The authors would like to thank E.N. Nigmatulina and M.V. Khlestov (IGM) for technical assistance at EMPA and SEM studies. We also would like to thank Adam Abersteiner (University of Tasmania) for assistance in the English correcting of the last version of the manuscript. We are highly appreciative of the valuable comments and suggestions of two anonymous reviewers.

**Conflicts of Interest:** The authors declare no conflicts of interest.

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
