# Peer review of "Uakitite, VN, a New Mononitride Mineral from Uakit Iron Meteorite (IIAB)"

_minerals, doi:10.3390/min10020150_

Round 1

Reviewer 1 Report

It is a nice discovery. The paper is well written and needs only minor edits.

Figure 1. Insert arrows to link labels to the two nodules.

Figure 2. The lower photos may be improved.

L 143. The Mo-phase can be identified using SEM-EDS-EBSD.

L 170. Fe-rich kalininite should be called Zn-rich daubreelite, according to its formula in Table 1, 

Figure 5. Insert arrows to link labels "VN-1" and "VN-2" to the mineral.

L 219-220. Re-write this sentence to make it clear.

L 220. n=52 should be n=54, based on Table 2.

L 265. Delete the sentence.

L 299. Change "cosmic" to "extraterrestrial".

Author Response

Thank you very much for your review.

Reviewer 1

It is a nice discovery. The paper is well written and needs only minor edits.

Figure 1. Insert arrows to link labels to the two nodules.

It is done.

Figure 2. The lower photos may be improved.

We deleted them and added new photos.

L 143. The Mo-phase can be identified using SEM-EDS-EBSD.

Unfortunately, it was not possible to indentify this phase using SEM-EDS and mapping due to very small size (< 0.5 µm). EBSD was not provided for this phase. Anyway, we added the size of the phase in the text.

L 170. Fe-rich kalininite should be called Zn-rich daubreelite, according to its formula in Table 1.

We do not agree here. The average formula presented in Table 1 is (Zn0.52Fe2+0.48)(Cr1.95Fe3+0.05)O4. The result is Zn>Fe2+.

Figure 5. Insert arrows to link labels "VN-1" and "VN-2" to the mineral.

It is done.

L 219-220. Re-write this sentence to make it clear.

It is done.

L 220. n=52 should be n=54, based on Table 2.

It is done.

L 265. Delete the sentence.

L 299. Change "cosmic" to "extraterrestrial".

It is done.

Reviewer 2 Report

The work reports the new mineral uakitite VN recently found in the uakit meteorite. The work is very well written and the science sounds good. All methodologies used are suitable for the mineral characterization and all literature is well quoted. So, there is not much for me to improve this work as I could not write it better than what it is. I therefore recommend to publish it as is with just very small typos that I report here below. From a crystallographic point of view, the authors could not do much more than what was done and thus the CIF file provided of course cannot satisfy the CheckCif test (the CIF provided by the authors is only based on calculated data, fine with me).

Line 220. Please I found a Cyrillic symbol between Cr and Fe.

Line 251. Please remove the symbol ° if the temperature is given in K.

Line 253. Here the temperature is given in °C.

Author Response

Thank you very much for your review.

Reviewer 2

The work reports the new mineral uakitite VN recently found in the uakit meteorite. The work is very well written and the science sounds good. All methodologies used are suitable for the mineral characterization and all literature is well quoted. So, there is not much for me to improve this work as I could not write it better than what it is. I therefore recommend to publish it as is with just very small typos that I report here below. From a crystallographic point of view, the authors could not do much more than what was done and thus the CIF file provided of course cannot satisfy the CheckCif test (the CIF provided by the authors is only based on calculated data, fine with me).

Line 220. Please I found a Cyrillic symbol between Cr and Fe.

It is changed.

Line 251. Please remove the symbol ° if the temperature is given in K.

It is done.

Line 253. Here the temperature is given in °C.

It is done.